# Improving Environmental Capacities for Health Promotion in Support Settings for People with Intellectual Disabilities: Inclusive Design of the DIHASID Tool

**DOI:** 10.3390/ijerph17030794

**Published:** 2020-01-28

**Authors:** Kristel Vlot-van Anrooij, Thessa I.M. Hilgenkamp, Geraline L. Leusink, Anneke van der Cruijsen, Henk Jansen, Jenneken Naaldenberg, Koos van der Velden

**Affiliations:** 1Department of Primary and Community Care, Intellectual Disabilities and Health, Radboud Institute for Health Sciences, Radboud University Medical Center, P.O box 9101, 6500 HB Nijmegen, The Netherlands; geraline.leusink@radboudumc.nl (G.L.L.); anneke.vandercruijsen@radboudumc.nl (A.v.d.C.); Henk.jansen@radboudumc.nl (H.J.); Jenneken.Naaldenberg@radboudumc.nl (J.N.); 2Department of Physical Therapy, University of Nevada, Las Vegas, NV 89154-3029, USA; thessa.hilgenkamp@unlv.edu; 3Department of General Practice, Intellectual Disability Medicine, Erasmus University Medical Center, Rotterdam, P.O box 2040, 3000 CA Rotterdam, The Netherlands; 4Department of Primary and Community Care, Radboud Institute for Health Sciences, Radboud University Medical Center, P.O box 9101, 6500 HB Nijmegen, The Netherlands; koos.vandervelden@radboudumc.nl

**Keywords:** health promotion, lifestyle, settings approach, health assets, intellectual disability, community participation, inclusive research, context-based interventions, empowerment, inclusion

## Abstract

People with intellectual disabilities (ID) have unhealthier lifestyles than the general population. To sustainably improve their lifestyle and health status, a whole-system approach to creating healthy environments is crucial. To gain insight into how support for physical activity and healthy nutrition can be embedded in a setting, asset mapping can be helpful. Asset mapping involves creating a bottom–up overview of promoting and protective factors for health. However, there is no asset mapping tool available for ID support settings. This study aims to develop an asset mapping tool in collaboration with people with ID to gain insight into assets for healthy nutrition and physical activity in such settings. The tool is based on previous research and development continued in an iterative and inclusive process in order to create a clear, comprehensive, and usable tool. Expert interviews (*n* = 7), interviews with end-users (*n* = 7), and pilot testing (*n* = 16) were conducted to refine the tool. Pilot participants perceived the tool as helpful in pinpointing perceived assets and in prompting ideas on how to create inclusive environments with support for physical activity and healthy nutrition. This overview of assets can be helpful for mobilizing assets and building the health-promoting capacities of ID support settings.

## 1. Introduction

People with intellectual disabilities (ID) have unhealthier lifestyles than people without disabilities, with more physical inactivity and unhealthy dietary habits [1,2,3,4], and their lifestyles contribute to many of their health problems [1,5]. The promotion of physical activity and healthy nutrition may help to decrease the health inequities faced by people with ID. However, people with ID are more dependent on their environment to live healthily. In a previous study on health promotion, people with ID expressed the need for a supportive social and physical environment to be able to live healthily [6]. This is supported by the growing evidence of environmental factors associated with lifestyle, such as the association between the presence of convenience stores and fast-food restaurants and nutrition intake, and the association between the accessibility of facilities, street safety, aesthetic attributes, and physical activity [7,8,9,10]. ID support settings are specialized in providing long-term residential, community living arrangements, and day activities for people with ID, who face limitations in intellectual functioning and adaptive behavior [11]. In the Netherlands, about 68,000 people live in facilities from ID support settings, ranging from clustered group homes to small-group living in apartments or single-family homes in neighborhoods [12,13]. People with ID spend a lot of time in these settings where they receive support with personal, daily, social, and home health tasks, mainly provided by daily care professionals trained in behavior aspects and/or assisted nursing [14]. So, environmental support for health promotion could contribute to sustainable improvement in the health status of people with ID and achieve more equality for this population in which ID support settings can play a crucial role. 

Despite the efforts of ID support organizations to improve the lifestyle of people with ID, the sustainable embedment of health promotion in daily support faces challenges [6,15,16]. On the one hand, many interventions developed by researchers in program settings are challenged by difficulties in implementing them in practice [15]. On the other hand, many of the interventions developed in practice focus mostly on the individual, consist of stand-alone activities, and lack embedment in policy [17,18]. Moreover, they lack sustainability as they are not embedded in the daily support system of ID support organizations [17,18]. To sustainably improve the lifestyles of people with ID in settings where they engage, a whole-system approach has been identified as a way forward [19].

Taking a whole-system approach is complex, as it requires health promotion to be embedded in the day-to-day practices of ID care organizations. This whole-system approach has been successfully implemented using the healthy settings approach [20,21]. This healthy settings approach is a whole-system approach where stakeholders are given the capacity to address behavioral and environmental factors and embed health within the routines and the culture of a setting [20,21]. It has been successfully implemented in hospitals and schools as healthy school and healthy hospital projects. These have resulted in transformed policies, organizational structures, and community action to facilitate healthy living [22,23]. Due to these successes in other settings, this approach might also be beneficial for health promotion in ID support settings. 

Co-creating healthy settings is key, as the people who actually use a setting know best which of the existing resources can be useful and how health promotion can be made part of the whole-system in a certain setting [24]. Asset mapping is a bottom–up process for creating an overview of those resources (promoting and protective factors) that maintain and sustain health and well-being in a defined setting. In this approach, people who use the setting are actively involved, as they have essential knowledge and experiences about living in a place and the resources available. Therefore, asset mapping can be used to provide input for the whole-system approach. In general, there is a lack of asset mapping techniques [25]. Although existing tools can help assess resources for health promotion in the environment [26,27,28,29,30,31,32,33,34,35,36], these tools fit poorly with an assets mapping approach due to the lack of a whole-system focus, the lack of a positive approach, or a scope that is too narrow. Furthermore, making a tool that can be used by people with ID themselves requires a clear structure and language with instructions to create meaningful engagement by people with ID.

This study aimed to develop a comprehensive, clear, and usable inclusive tool for environmental asset mapping for ID support settings that can also be used by people with ID themselves. The tool, developed in Dutch, provides insight into perceived environmental assets and points for improvements regarding support for healthy nutrition and physical activity for people with moderate to profound ID in settings where they engage. These insights can be used to create inclusive and health-promoting environments. This article describes the iterative and inclusive development process of creating the tool using expert interviews, cognitive interviews, and pilot testing. The inclusive research team used this input to create a functional tool that can be used by people with ID and care professionals. 

## 2. Materials and Methods 

### 2.1. Development Phases

This study used an iterative process in which end-users were involved to develop the asset mapping tool named DIscovering Health-promoting Assets in Settings for people with Intellectual Disabilities (DIHASID). End-users include people who engage in a living or day-activity location, e.g., people with mild to moderate ID, proxy respondents for people with severe/profound ID, and care professionals. The three development steps are visualized in Table 1 and described below. 

The DIHASID tool is underpinned by an ecological model and the theory of salutogenesis. This implicates a focus on multiple environmental levels and on protective or promotive factors rather than on barriers and needs, and a focus on assets [37,38].

The research team developed a draft asset mapping tool based on the Healthy Settings for People with Intellectual Disabilities (HeSPID) framework. This framework was built on two studies in which academics, people with ID, and proxies for people with ID developed a framework of themes and assets relevant for physical activity and nutrition in ID support settings [39,40]. The framework consists of 14 environmental-asset themes relating to people, places, and preconditions for healthy living. 

This draft was discussed during expert interviews. Focus points were elicited on the comprehensibility of the themes and questions of the DIHASID tool, including all possible assets relevant for healthy living in ID support settings. Firstly, the aim of the DIHASID tool and an overview of the themes were introduced. Secondly, each theme was introduced, with a description based on the HeSPID framework [39,40]. Then, for each theme, the questions were read aloud, and the participants were asked to provide feedback on how representative the questions were. In addition, further suggestions were requested. Lastly, participants were asked to reflect on the tool and share ideas on other themes that should be included. 

The cognitive interviewing (CI) technique was used to check the clarity of the questions for the users. CI is a method to evaluate the quality of transferring knowledge in questionnaires and has been used successfully among people with ID [41,42]. In CI, the interviewer reads the questions aloud and asks the interviewee to think aloud when answering the question. Probing questions are used to let the interviewees paraphrase questions, discuss thoughts, feelings, and ideas, and suggest alternative wording. The Question Appraisal System (QAS-99) was used to develop the interview protocol, including probing questions related to possible problems identified by the research team [41]. The interviews started with an explanation of the aim of the interview and the tool. Then, each question was read aloud by the interviewer, and the interviewee expressed what he/she thought and what he/she would answer. If applicable, probing questions related to the question were asked. After one hour, the interview stopped, unless the interviewee explicitly wanted to continue. Interviews were audiotaped and conducted by K.V.v.A. in a place that was convenient for the interviewee. 

To improve the usability of the DIHASID tool in practice, people with ID and care professionals at the three pilot locations (1) completed the DIHASID tool; (2) completed the After-Scenario Questionnaire (ASQ), a 3-item questionnaire about user satisfaction [43]; and (3) participated in a group discussion in which task usability, user satisfaction, functional usefulness, and ideas for improvements of the DIHASID tool were discussed. The group discussion topics were based on usability domains [44]. 

### 2.2. Procedures

For the expert interviews, experts were sought on physical activity, nutrition, and health promotion for people with ID. For the cognitive interviews, end-users were recruited: adults with mild/moderate ID who are able to communicate verbally, proxy respondents for persons with severe/profound ID, and a care professional. Diversity was sought in type of location (living or day-activity location). For the pilot, living or day-activity locations for people with moderate to profound ID were sought. In each pilot location, between two and four care professionals and between two and four adults with mild/moderate ID who were able to communicate verbally or between two and four proxy respondents for adults with severe/profound ID were recruited. Participants were recruited through purposive sampling. For the expert interviews, the research team’s network was used to recruit participants by inviting them through email. For the cognitive interviews and pilot, the contact persons of eight ID support providers helped to recruit participants. They sent the information leaflet to team leaders and care professionals and asked them to identify potential participants.

The care professionals identified potential participants who were interested and able to participate and provide consent. Care professionals provided them with an information leaflet on the content and procedure of the study. If needed, the care professionals assisted in reading and understanding the information. Those who were interested to participate were asked to read or listen to the consent form. It was possible to contact the researcher by phone or email to ask questions. Those who agreed to participate were asked to sign the form themselves. After consent was obtained, the contact information was shared with the researcher, who contacted them or their care professional to schedule the meeting(s). For the expert interviews, informed consent was obtained when the appointment was being made.

The study was conducted according to the principles of the Declaration of Helsinki and the EU General Data Protection Regulation. The Medical Research Ethics Committee of Radboud University and Medical Center approved this study (registration number: 2018-4408).

### 2.3. Inclusive Approach 

This study actively involved people with ID as co-researchers to deploy experiential and scientific knowledge and contribute to appropriate data collection, data quality, and relevant outcomes [45,46]. The inclusive research team consisted of researchers with ID (co-researchers) and without ID, all employed by the university, and followed Frankena’s [46] guidelines in the consensus statement for inclusive health research. K.V.v.A., A.v.d.C., and H.J. developed the procedure and the data collection method and incorporated feedback from other team members and the project’s advisory group, which included people with ID, care professionals, health professionals, and a manager. Data collection and analysis was conducted by K.V.v.A. The co-researchers assisted when interpretation questions arose regarding the analysis of the cognitive interviews and group discussions of the pilot. Then, they listened to the audiotapes and discussed the meaning of what participants said. After each phase, K.V.v.A., A.v.d.C., H.J., and J.N. discussed how to adjust the tool in light of the problems and possible solutions identified during data collection. Given the important contribution of the co-researchers to this study, they are also recognized as co-authors on this paper. 

Collaboration between the researchers with and without ID was supported by (1) the research clock, a clock on which steps of the study were visualized to prompt memory; (2) audio recordings rather than transcripts for data analysis; (3) verbal explanation of this manuscript to obtain feedback; and (4) a training on working as a team of researchers with and without ID. In addition to this scientific paper, an easy-read abstract was written.

### 2.4. Analysis 

Data analysis was performed using Atlas.ti software 8.2.29 and SPSS (version 25, SPSS Inc., Chicago, IL, USA. The suggestions from the expert interviews were collected and grouped based on type of problem and suggested improvements for the DIHASID tool. The audio recordings of the cognitive interviews were analyzed using Atlas.ti. The identified problems were selected and categorized according to the eight QAS-99 categories [41]. Then, the categories were thematically analyzed, and suggestions for improvements were logged. 

The pilot data on the DIHASID tool and the ASQ were analyzed using descriptive statistics in SPSS. The audio recordings of the group discussions were thematically analyzed using Atlas.ti. Relevant fragments were structured in the categories of the TURF framework on usability, where TURF stands for Task, User, Representation and Function [44], and then thematically analyzed. The gathered information was discussed among the research team to finalize the DIHASID tool.

## 3. Results

### 3.1. Participants

Thirty persons participated in the development of the DIHASID tool. Seven female experts in lifestyle and health promotion participated in interviews on the comprehensibility of the DIHASID tool: three experts on physical activity for people with ID, two experts on nutrition for people with ID, and two experts on health promotion. The following end-users participated in cognitive interviews on the clarity of the DIHASID tool: people with ID, aged 18–55 (two male, three females, three filled it out for living location and two for day-activity location), a female proxy respondent (parent), and a female care professional. The tool was piloted on usability among 16 persons from three different living and/or day-activity locations for people with moderate to profound ID, i.e., six persons with ID (five males, 1 female), two female proxy respondents, seven female care professionals, and one male manager.

### 3.2. Comprehensive DIHASID Tool

The analysis of the expert interviews resulted in six points for improving the comprehensibility of the DIHASID tool: (i) add a theme, (ii) add answer options, (iii) clarify or divide broad or vague questions, (iv) find better matching response categories for which respondents have the knowledge to answer, (v) use reminders for what is viewed as healthy living and a healthy living environment, and (vi) personalize questions. The input was used to change the tool regarding (i) adding or changing questions and answer options, (ii) providing more instructions, and (iii) personalization of the questions. Table 2 provides a full list of the points for improvements and changes made to the DIHASID tool.

Analysis also resulted in points that did not match the aim of the DIHASID tool and therefore did not result in changes to the tool. Examples include suggestions on the knowledge or professional attitude of clients and care professionals, relaxation, and negative environmental factors. The stability of the social network of people with ID was not included in the DIHASID tool, as this was perceived as too difficult to ascertain via a questionnaire. Details on accessibility (e.g., does the swimming pool have a hoist) were not included, as this would make the list too detailed and too long.

### 3.3. Clarity of the DIHASID Tool

Analysis of the cognitive interviews identified 152 problems with clarity, resulting in 119 adjustments to the DIHASID tool. The problems and their adjustments are described below using the eight QAS-99 categories, see Table 3.

In the Clarity category (*n* = 64), problems related to the wording of the questions, technical terms, such as health professionals and epilepsy, and vague questions. Regarding Response categories (*n* = 38), problems related to technical terms and vague, overlapping, and missing answer options. For example, the differences between the five smileys were vague according to the participants. Problems with Instructions (*n* = 23) included lack of clarity on what to consider when answering the questions, information missing on how many answers could be chosen, and surplus information. A few problems related to Knowledge or Memory (*n* = 10), including difficulty in knowing the boundaries and facilities of—and distances from—facilities within the neighborhood and care professionals’ knowledge on organizational policy and budgets. For Sensitivity or Bias (*n* = 7), problems related to questions on the nature of a person’s disabilities and use of the word ‘client’. Only one problem related to the Assumptions category: it was perceived as difficult to choose one smiley for how a person perceives help from all health professionals. Other problems (*n* = 9) related to the size and unclear meaning of pictures.

The identified problems and suggestions were used to improve the clarity of the DIHASID tool by shortening and specifying instructions, explaining how many answer options to choose and where to fill in the answer, including or changing pictograms, changing word order, replacing technical terms with easy words, explaining unclear words, removing/inserting answer options, and changing sensitive words. 

### 3.4. Usable DIHASID Tool

The analysis of the DIHASID tool pilot provided information on (1) how the task was performed and experienced, (2) final points for improvements on usability, and (3) what the DIHASID tool can yield in practice.

It took the 16 participants on average 34 min (38 for participants with ID, 35 for proxy respondents, and 30 for care professionals) to complete the task, and only a few answers were missing. Seven participants, of which six people with ID, chose to fill the DIHASID tool out on paper, and nine used the online questionnaire; both were perceived as clear and easy to navigate. Most participants perceived the explanation and clarity of the task (*n* = 13 out of 16), the ease of the task (*n* = 12 out of 16), and the length of the task (*n* = 13 out of 16) as good. All participants viewed themselves as the right person to answers the questions, except those on financial aspects and health-promoting organizational policies, which care professionals perceived as difficult because they were not familiar with these issues. Regarding financial and policy aspects, participants identified a team leader as the right person to be involved in filling out the DIHASID tool. Participants with ID perceived the help from a care professional as pleasant, needed, and not influencing their answers.

Final points for improving the DIHASID tool included: (1) page numbering, larger answer fields, and larger fonts for the paper version, (2) allowing participants to choose more than one answer option for multiple choice questions, (3) instructing proxies that they can tick ‘not applicable’ for questions that are irrelevant for the person they represent, e.g., a question about talking when the person they represent cannot speak, and (4) final changes to questions and explanations to improve clarity, for example changing the description of clients ‘resident or participant at daytime activities’ back to ‘client’.

In the group discussions, participants reflected that the DIHASID tool can help to (1) raise awareness and put healthy living in the spotlight, (2) create an overview on what is available to support healthy living, and (3) use the overview to create changes in the organization. Participants identified a summary of the outcomes as needed for generating actionable knowledge. For example, teams of care professionals can discuss this summary and devise action steps together. Participants identified the following stakeholders with whom to share this summary: clients, clients’ families, care professionals, team leaders, personal support coordinators, policymakers, and quality assurance officers of the organization.

### 3.5. Final Version of the DIHASID Tool

The final DIHASID tool (see Appendix A) consists of 37 questions divided into four parts: (1) participant and setting characteristics, (2) how people support healthy living including their social network, types of support, and values regarding healthy living, (3) how places support healthy living including tools, facilities, accessibility, and person–environment fit, and (4) the preconditions for healthy living that are available, including financial aspects and health-promoting organizational policies. Regarding the type of questions, part one includes multiple choice questions. Parts two, three, and four include the following type of questions: (1) tick boxes on presence of assets, (2) multiple choice questions (3-point smiley scale, but 5-point Likert scale for questions that are aimed only at care professionals and proxies) on how respondents experience a theme, and (3) an open question on wishes and dreams regarding the theme. The tool can be completed by people in a living or day-activity location, e.g., people with mild to moderate ID, proxy respondents for people with severe/profound ID, care professionals, and team leaders.

## 4. Discussion

This study aimed to develop an inclusive and functional tool for mapping assets for physical activity and healthy nutrition in ID support settings. An iterative process of applying feedback from expert interviews, cognitive interviews, and pilot testing was used to develop a comprehensive, clear, usable tool. The tool, named DIscovering Health-promoting Assets in Settings for people with Intellectual Disabilities (DIHASID), can be completed in approximately 30 min by people with mild to moderate ID who are assisted by a support person, proxy respondents for people with severe/profound ID, care professionals, and team leaders.

The DIHASID tool is an inclusive tool for people with ID and care professionals that can be used to facilitate bottom–up engagement to improve the health-promoting capacities of ID support settings. This approach is empowering and aligns with the ‘Nothing about us, without us’ movement that advocates for the involvement of people with ID in matters that affect them [47]. Furthermore, this bottom–up approach can create awareness among policymakers of what supports people with ID and their care professionals in facilitating healthy lifestyles. The DIHASID tool helps to implement inclusive and healthy environments and thereby facilitates policymakers in the trend toward a greater focus on environmental impacts on health. For example, the Dutch Environment Act and Green Deal provide good opportunities to include attention on health promotion in spatial planning and sustainable innovations, including a healthy living environment in the care sector [48,49].

Participants perceived the DIHASID tool as helpful for providing an overview of user-experienced assets and wishes regarding a healthy living environment for physical activity and healthy nutrition of people with ID, thereby aligning with the goals of asset mapping [24]. From an asset-based community development perspective, the next steps for building healthy ID support settings include (1) finding connectors and engaging them in (re)building relationships between people to link assets and create a health-promoting infrastructure, (2) creating a joint vision and action plan, and (3) embedding this plan and vision in the settings’ organizational structure [50,51]. These steps are important but also challenging to implement in ID support settings because currently there is a lack of clarity among stakeholders on roles and responsibilities regarding health promotion. Care professionals who are involved in everyday support are often not trained on this topic. Allied health professionals often focus mostly on curative care rather than prevention and may not know how to facilitate care professionals [19,52]. Furthermore, it might be challenging to involve people with ID in developing a joint vision and action plan. Future studies could design and pilot how this bottom–up process can be tailored to their needs.

A major strength of this study is the co-design of the DIHASID tool by the inclusive research team together with experts in practice, experts in research, and experts by experience. This ensured that tailored methods were used to enable people with ID to meaningfully engage as participants and led to a better match between research and practice. In addition, the insights of the researchers with ID helped in interpreting user perspectives and in deciding on appropriate changes to improve the usability of the tool.

The number of interviews to improve the comprehensibility, clarity, and usability of the tool was limited. However, an iterative process was used, and after the pilot, hardly any changes were required. Although the DIHASID tool gives prompts about a wide range of assets in the physical, social, and organizational environment, the results depend on the participants’ familiarity with local assets. Therefore, it is preferable that multiple persons in a setting fill out the DIHASID tool to gain an overview that is as complete as possible. Lastly, some caution should be exercised about implementing this tool in other countries. The type of questioning and general themes are expected to be relevant in other countries, as the tool was built on the basis of an existing international concept mapping study [39]. However, the clarity of the questions was tested in Dutch, and the tool’s comprehensibility and usability were tested in the support organization in the Netherlands. Therefore, we advise anyone who wants to apply the DIHASID tool in another country to conduct a pilot to see whether adaptations are needed for that context.

Future studies could use the DIHASID tool to (1) provide insight into how people with ID are currently supported by ID support organizations to live healthily, (2) enhance intervention effectiveness in specific settings by identifying assets that can support the intervention in that particular setting and/or interweave the intervention in the setting [53], and (3) gain insight into contextual factors that might influence the outcomes and successes of health promotion interventions applied in that particular setting [54].

## 5. Conclusions

The DIHASID tool is a comprehensive, clear, and usable tool to map health-promoting assets in ID support settings. Using the tool provides insight into perceived environmental assets and into points for improvements regarding support for healthy nutrition and physical activity of people with moderate to profound ID in settings where they engage. The bottom–up development of this tool for co-learning ensures that the DIHASID tool asks about assets that may be relevant for users of ID support settings. The tool empowers people with ID and care professionals to pinpoint assets that they find helpful and to identify future directions for creating healthy environments for physical activity and healthy nutrition. The tool can be used together with stakeholders who are responsible for health promotion and organizational policy, and the overview of assets can be used to mobilize and build on assets to inclusively improve the health-promoting capacity of ID support settings.

## Figures and Tables

**Table 1 ijerph-17-00794-t001:** Development of the DIHASID tool: phases, action, results, and participants. ID: intellectual disabilities.

Phase	Action	Result	Participants
Make the DIHASID tool comprehensive	Check the extent to which the DIHASID tool represents all facets of a given construct	Based on expert feedback, the DIHASID tool is adjusted to make it comprehensive	Experts on physical activity, nutrition, and health promotion for people with ID (*n* = 7)
Make the DIHASID tool clear	Check the readability, clarity of language, and consistency of style of the questions and format of the DIHASID tool	Points of attention deduced in the cognitive interviews are used to improve the clarity of the DIHASID tool	End-users: people with mild/moderate ID, proxy respondents for people with severe/profound ID, and care professionals (*n* = 7)
Make the DIHASID tool usable	Pilot test the DIHASID tool to test the usability of the scan in settings where people with ID live, work, and engage	Pilot testing improves the tool’s usability, and the final DIHASID tool is developed	End-users from three pilot locations (*n* = 16)

**Table 2 ijerph-17-00794-t002:** Points for improvement suggested in expert interviews and changes to the DIHASID tool.

Point for Improvement	Changes to the DIHASID Tool
**Add theme:** Include communication about healthy living within an organization in questions about health-promoting organizational policies.	The question: “How do you perceive the attention on healthy living in communications by this organization?” was added.
**Add answer options for the questions:** (1) Type of disabilities: type of wheelchair, I am not allowed on the road by myself, epilepsy (2) Type of support persons: friends, occupational therapist, speech therapist (3) Type of support: others buy food/devices: bicycle for the wheelchair, book with ideas about exercise activities, games in which you need to move, meal service, and meal-in-a-box (4) Type of autonomy-supported decision making: clients choose themselves, they do not receive help.	The suggested answer options were added to the questions.
**Clarify or divide broad or vague questions:** (1) The answer options for the question on types of advice from types of health professionals are not complete. Many health professionals can give several types of advice. (2) How participants experience the help of others for healthy living is very broad. It might be better to split ‘others’ into categories such as family and friends, health professionals, care professionals, volunteers, and clients. (3) The question, “What do you think of the opportunities for healthy living in the neighborhood?” was found to be vague. This could be interpreted as places for healthy living or activities for healthy living.	(1) The question was split into two questions: “At this location, there is enough opportunity for care professionals to get tips about...?” <answer options include types of advice> and “Who is available to provide this advice?” <answer options include types of health professionals>. (2) The answer option for the question, “How well do others help with healthy living?” was split into three categories: (a) care professionals, clients, and volunteers, (b) family and friends, and (c) health professionals. (3) The question was split into: “Are there enough places for healthy eating, healthy drinking, physical activity, and sports in the neighborhood?” and “Are there enough activities for healthy living in which you/the client can participate?”
**Matching response categories:** (1) The answer type for the question on talking about healthy living was perceived as difficult and not appropriate. The answer type on how often talks about healthy living were held was perceived as less important than how talking is experienced. (2) The answer option for the questions, “How much time do care professionals have for activating clients?” and “How much time and attention and do care professionals have for providing food and drinks?” were perceived as too difficult. It was perceived as too difficult for participants to express this in days per week, as this largely varies between weeks.	(1) The answer options were changed to a 5-point smiley answer. (2) The answer options were changed to never/sometimes/often/always.
**Use reminders:** The experts stated that clients would need reminders of what is viewed as healthy living and a healthy living environment.	The explanation of healthy living was repeated at several places in the questionnaire. The subthemes of People, Places, and Preconditions were repeated above the open questions to stimulate the participants to think about all the questions that they answered about the overarching theme and formulate wishes.
**Personalized questions:** The participants perceived referrals in questions as too general. Personalization of the questions was perceived as helpful for clients (e.g., ”Who supports you with healthy living?” instead of “Who at this location supports healthy living?”).	Separate questions were devised for clients, proxies, and care professionals.

**Table 3 ijerph-17-00794-t003:** Problems identified in cognitive interviews and changes to improve the clarity of the DIHASID tool.

QAS-99 Category	Description of Problems	Changes to the DIHASID Tool
**1. Reading** Difficulty reading the question (what and how to read)	n.a.	n.a.
**2. Instructions** Problems with instructions or explanations (conflicting, inaccurate, or complicated)	-unclear for participants what to consider when answering the questions -unclear instruction on the number of answers that can be chosen -unclear what to write or where to write an answer -difficult explanations: pictograms with words under them would help them understand the question better -some information was perceived as surplus -including that a support person is allowed to help was perceived as helpful for getting answers to the open questions	-shorten the questionnaire instruction -specify the instruction -explain how many answers may be chosen-specify that help from a support person is allowed -explain where to fill in the answer -include pictures and words beneath them
**3. Clarity** Problems related to communicating the intent of the question (wording, technical terms, vague, reference points)	-participants had difficulty understanding the sentence for some questions -technical terms, such as health professionals, aids, patient lift, masseur, epilepsy, spasm, residential and daytime support center -vague questions, for example what a neighborhood is	-change word order in sentences -give explanation or examples for unclear words -replace technical terms with easy words
**4. Assumptions** Problems with assumptions made or underlying logic (inappropriate, assumes constant behavior, double-barreled)	-it was perceived as difficult to choose one smiley for how a person perceives help from all health professionals	n.a.
**5. Knowledge/Memory** Whether respondents are likely to know or remember information (knowledge, attitude, recall failure, computation problems)	-difficulty in knowing the boundaries and facilities of, and distances from, facilities within the neighborhood -for care professionals: to know about the policy and financial budget of their organization	-make the distance from facilities broader (within 15-min walking distance, within 15-min biking distance, you need a car/cab/bus to get there) -insert “I don’t know” options for questions for care professionals about budget and policy
**6. Sensitivity/Bias** Sensitive nature, wording, or bias of questions (sensitive content or wording and social acceptability)	-the nature of a person’s disabilities -use of the word client	- include the response option “I don’t want to say” for the question about disabilities -change client into resident or participant at daytime activities
**7. Response categories** Adequacy of range of responses (difficulty of open-ended questions, mismatch, technical terms, vague, overlapping, missing, illogical order)	-unclear technical terms: fitness center, hydrotherapy bath -vague answer options: smiley response categories because differences between the five smileys were unclear for participants -overlapping answer options: kitchen and adjusted kitchen -missing answer options: vegetable garden for the question about aids for healthy nutrition	-replace technical terms with easier words -change words or add examples for vague answer options -remove answer options (use of three instead of five smileys) -remove overlapping answer options -add open answer options for incomplete response categories
**8. Other problems**	-size of pictures -unclear meaning of pictures	-size of all pictures was increased -unclear pictures were changed into pictures that were perceived to be clearer

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
