# Peer review of "Improving Environmental Capacities for Health Promotion in Support Settings for People with Intellectual Disabilities: Inclusive Design of the DIHASID Tool"

_ijerph, 2020, doi:10.3390/ijerph17030794_

Round 1

Reviewer 1 Report

The manuscript by Vlot-van Anrooij et al. “Improving environmental capacities for health promotion in support settings for people with intellectual disabilities: Inclusive design of the DIHASID tool,” seeks to provide a tool to understand the status of healthy living environments and what can be done to improve those environments for people with intellectual disabilities (ID). The manuscript describes an iterative process that utilized caregivers and people with ID to develop and pilot test an asset mapping tool. The result of the study is a tool that may be used by a wide array of stakeholders to improve the healthy living environment of people with ID.

A few improvements may help others to understand more about process to develop and implement this tool.

What was the age range and/or intellectual ability (IQ) range of the people with ID who assisted in reviewing the DIHASID tool? Could you define these individuals with a little more clarity than “adults with mild/moderate ID who were able to communicate verbally?” (line 140) Please give more detail to the ways individuals with ID were protected during the informed consent for the ethics approval. How much autonomy was given to individuals with ID and what were the criteria for allowing them to consent for participation in the study? Does including them as “co-researchers” protect them as individuals? In line 203-204 it should read “does the swimming pool have a hoist.” The use of the word “client” seems to be a problem with individuals who reviewed the DIHASID tool. (line 223) Yet, “client” is still included in the tool, and was confusing to this reviewer. Is there a better term that could be used to replace “client?” The statement, “Care professionals who are involved in everyday support do not possess the necessary knowledge, skills, and power, and allied health professionals who possess these assets do not know how to facilitate care professionals” (lines 303-306) seems condescending and stereotypical to both care professionals and allied health professionals. There may be individuals within each of these groups that possess the attributes that the statement indicates they do not have. Instead of “do not possess” and “do not know how” perhaps “may not possess” and “may not know how” should be used to be more inclusive. In the Health Environment Survey when choices appear in brackets, it was confusing (e.g. [I am a/The person on whose behalf I am filling out the survey/The client group consists of]). Is there a better way to make these questions easier to understand?

Reviewer 2 Report

The article describes the development of the DIscovering Health-promoting Assets in Settings for people with Intellectual Disabilities (DIHASID) tool, which is a tool for measuring health assets in the setting in which people with mild intellectual disabilities live.  Overall the development of the tool is well described and the tool itself appears to be useful, at least in the Dutch context.  The article does not describe the implementation of the tool, and the tool itself is not tested, so it is not known how valid and reliable this tool is, nor whether it actually leads to improvements in health for people with disabilities.  Nevertheless the methodology for developing the tool, which employed a co-design approach, was interesting and innovative.  The article is well written with only a few typos and grammatical issues. The article could be somewhat improved by a clearer description of the sampling and recruitment processes, and also some of the issues and challenges faced in the co-design process.  Another issue is that there is an extensive table which describes changes made to the tool, but this is very hard to follow as the tool itself is not provided.  

Another minor issue is that it is only mentioned at the end of the article that the project was conducted in the Netherlands and that the findings may not be generalisable to other countries - or at least the tool will have to be translated.  This should be stated in the introduction, and perhaps some context provided about care settings for people with ID in the Netherlands.  
